# The Surface Modification of Papers Using Laser Processing towards Applications

**DOI:** 10.3390/ma16206691

**Published:** 2023-10-14

**Authors:** Mariusz Dudek, Elżbieta Sąsiadek-Andrzejczak, Malwina Jaszczak-Kuligowska, Bożena Rokita, Marek Kozicki

**Affiliations:** 1Institute of Materials Science and Engineering, Lodz University of Technology, 90-537 Lodz, Poland; 2Department of Mechanical Engineering, Informatics and Chemistry of Polymer Materials, Lodz University of Technology, 90-924 Lodz, Poland; 3Institute of Applied Radiation Chemistry, Lodz University of Technology, 90-924 Lodz, Poland

**Keywords:** laser marking, surface complexity, microprinting, paper packaging, decorative paper, tactile detection

## Abstract

This work presents the results of paper laser processing. It begins with the selection and examination of the processing parameters, then an examination of the properties of the modified papers and examples of applications of the developed modification method. The properties of laser-modified paper were studied using reflectance spectrophotometry to examine the colour aspects of the modified papers, scanning electron microscopy (SEM) and confocal microscopy for a morphological analysis, and Raman spectroscopy to analyse the papers under the influence of laser light. The influence of laser processing on the wettability of paper and the evenness of unprinted and printed paper was also investigated. The knowledge gained on paper surface modification with laser light was used to propose several applications, such as methods of marking, tactile detection, the controlled removal of optical brightener, ink, and metallised coatings from paper packaging, highlighting the design and aesthetics of paper. The developed laser-assisted method shows a promising, ecological approach to the design of many value-added paper products.

## 1. Introduction

There are many types of paper on the commercial market. Depending on its intended use, paper has different properties and structures. In the general classification, there are papers made of a pulp with a grammage of up to 225 g/m^2^ [1]. Papers can be made from various pulps that differ in their chemical composition, with mainly cellulose and lignin content. Pulp compositions are used to produce printing papers, which ensure the appropriate properties of the paper following the requirements of the printing technology. In addition to fibrous semi-products, bulk additives are also used in the production of paper: fillers, adhesives, dyes, pigments, and auxiliary chemicals. Moreover, in the production of white paper, white pigments such as kaolin, chalk, gypsum, talcum, calcium carbonate, or titanium oxide are added. These fillers increase the opacity, whiteness, softness, smoothness, and dimensional stability of the paper. The most common printing papers for office use are made from bleached, medium-filled, wood-free paper [2]. Papers for producing packaging can be additionally multilayered, varnished, and coated [3].

The parameters characterizing papers, such as grammage, thickness, volume, dimensions, smoothness, transparency, cohesiveness, mechanical properties, humidity, stiffness, optical properties, degree of whiteness, and colour, determine their use in producing various paper products and their refinement. In the paper industry, printing is the most popular method of paper refinement. Printing is a low-cost method that enhances the aesthetics of products or gives them special features, for example, in banknote production, where optically variable or phosphorescent inks are used for security against counterfeiting [4,5]. Recently, laser techniques have also been used for paper refinement and modification. The most common are laser ablation for paper cleaning from dust, dirt, or inks [6], micropattering for a wide variety of applications ranging from microfluidics to rapid medical diagnostics [7,8,9], cutting, and engraving. The laser-cutting process of wood-based papers is a thermochemical decomposition process. When the laser beam reaches the surface of the paper, it heats up the material to its evaporation temperature and causes the material to sublimate. The energy of the laser beam interacts with the paper, breaking the chemical bonds, thus disrupting its structure [10]. Paper cutting using the laser technique is most often used in printing processes as a modern method of confectioning paper products, which eliminates the use of expensive paper cutters. It gives the possibility of cutting non-standard patterns that can be easily and quickly prepared in vector graphics software. In terms of securing paper products against counterfeiting, laser engraving seems to be more interesting. This method allows for changing the tactile features and can be used for the laser marking of paper products such as banknotes and documents [11]. However, the research works presented in the literature do not exhaust the subject of surface paper modification, since most efforts have been focused on paper cutting, engraving, cleaning, and micropattering.

The aim of this work is to present the method of a laser-aided surface modification of paper and cartons, and is based on previous experience related to the use of lasers in surface engineering [12,13]. At first, the parameters of the paper treatment with laser were established. Afterwards, the most important parameter of laser power was investigated with respect to the obtained effects on paper surfaces. The effects on papers and cartons were examined for different surface masses (grammage) and colours. Several techniques were employed for the characterization of the modified papers: reflectance spectrophotometry for an analysis of the colour changes of the samples, a flat-bed scanner, scanning electron, and confocal microscopy for morphological analyses, Raman spectroscopy for a chemical analysis, and contact angle measurements for an investigation of the hydrophilicity of the samples. Finally, several examples of applications related to marking, microprinting, tactile detection, labelling, controlled paint, and optical brightener degradation, with an emphasis on the design and aesthetics of paper packaging, are also presented.

## 2. Materials and Methods

### 2.1. Laser Processing

The paper samples prepared from commercial paper used for office printers (POLspeed, Kwidzyn, Poland) were modified using a 20 W G3 SPI pulsed laser system (SPI Lasers UK Ltd., Southampton, UK) with a wavelength of 1064 nm and a line band width of <4 nm. The paper properties are presented in Table 1.

The laser beam was delivered to an XLR8 2-axis scan head (THORLABS (Nutfield Technology), Hudson, NH, USA) and F-Theta 160 mm focusing lens. The scanner head with the F-Theta lens allowed for moving the focused laser beam (with a spot of about 40 μm) over the substrate, keeping the focal position on the substrate surface. A hatching technique was used on the machine over a paper surface. The technique consists of filling a defined shape with lines. Preliminary experiments established the initial conditions for the laser modification. Thus, the paper samples were modified using the following parameters: (i) average power: 2–20 W, (ii) pulse frequency: 35 kHz, (iii) pulse duration: 220 ns, (iv) scanning speed: 1200 mm/s, (v) hatching distance between parallel lines: 35 μm, and (vi) always bidirectional hatching direction with two angles: 0° and 90°. To check the influence of the grammage and colour of the paper on the modification effect, white paper samples with grammages of 190 and 250 g/m^2^, as well as printed papers with various colours (yellow, red, green, blue, and black), using an Epson L300 printer (Epson, Suva, Nagano, Japan—original water-based inks were used), and coloured cartons were also subjected to laser treatment (POLspeed, Kwidzyn, Poland).

After the laser processing, all the paper samples were documented with a Canon EOS 50D (Canon, Tokyo, Japan) digital camera under D65 illuminance. The photos presented in the manuscript were not colour-enhanced.

### 2.2. Reflectance Measurements

The reflectance spectra of the paper samples were measured using a Spectraflash light reflectance instrument (Spectraflash 300, D65/10° (where 10 describes an angle of illumination); 10 nm step, the measurement error was 0.1%, DataColor, Rotkreuz, Switzerland). The samples were measured over the wavelength region of 400–700 nm. Chromatic colours [14] are described by using the *L***a***b** description, where *L** represents lightness from black to white on a scale of zero to 100, while the value “*a**” represents the green–red axis and the value “*b**” represents the blue–yellow axis.

### 2.3. Scanning Electron Microscopy Measurements

The morphology of the paper samples was analysed with a TESCAN VEGA3–EasyProbe (TESCAN Brno, s.r.o., Brno, Czech Republic) scanning electron microscope equipped with a VEGATG software version 4.2.4.0 (high-vacuum mode (SE); accelerating voltage 10 kV). Before the measurements, the samples were coated with Au-Pd layers using a Cressington Sputter Coater 108 auto system (Cressington Scientific Instruments Ltd., Watford, UK). 

### 2.4. Confocal Microscopy Measurements

The treated surface of the papers was examined using a confocal laser scanning microscope (CLSM) Nikon MA200 (Nikon, Tokyo, Japan). The surface maps obtained using the CLSM were analysed using the MountainsLab^®^ 5.0 software package (Digital Surf, Besançon, France) to determine the roughness and complexity of the treated surfaces. Complexity describes how much larger the actual surface of the sample is than its geometrical dimensions:(1)$$\mathrm{complexity}=\left(\frac{\mathrm{actual}\;\mathrm{surface}\;}{\mathrm{geometrical}\;\mathrm{dimensions}}-1\right)\cdot 100\%,$$

Based on previous observations [13], complexity better reflects the surface modification process using a pulse laser than any of the roughness parameters.

### 2.5. Raman Spectroscopy Measurements

The chemical structure of the laser-processed paper was investigated using a Raman spectrometer (inVia Renishaw, Gloucestershire, UK), equipped with a 785 nm laser arranged in backscattering geometry. The investigated wavenumber ranged from 100 to 3200 cm^−1^. The spectra were collected at selected points and along a line crossing the borders between the functionalized and non-functionalized zones of the sample. The distance between the points during profile measurements was 10 µm. All the measurements were carried out in the air at room temperature.

### 2.6. Wettability Measurements

The water contact angle and the water drop absorption time were measured for the unmodified and laser-modified paper samples. The measurements were made with a Contact Angle Analyser (Phoenix Alpha, SEO, Suwon, Republic of Korea). The contact angle was calculated with the ImageJ programme (NIH, Bethesda, MD, USA). The surface wettability was assessed using Young’s equation. The drop absorption time was counted with a stopwatch from the moment the drop was placed on the surface until the drop was completely absorbed. During the measurement, the place of the water drop absorption was observed with a camera.

### 2.7. Evaluation of the Surface Unevenness

To determine the unevenness of the paper surface, firstly, the samples were scanned with an Epson Perfection V750 Pro scanner (Nagano, Japan; cold cathode fluorescent lamp; optical resolution Main 6400 DPI × Sub 9600 dpi; 48 bit/colour). The scanning was performed in reflection mode, and the colour depth was 24-bit RGB. All the samples (30 × 30 mm^2^) were scanned at a resolution of 300 dpi. Other scanning parameters (brightness, colour saturation correction, colour regulation, and sharpness) were switched off and not considered in this study. Based on the scans, calculations of the sample profiles were performed using the prepared script for reading RGB channels (RGBreader; Python Script with Python Imaging Library; DosLab [15]). Each sample was depicted using a three-colour RGB scale (red, green, and blue). After a preliminary analysis of the examined samples, the red channel was chosen for all the samples for which the highest changes in values were observed (from 0 to 255).

## 3. Results and Discussion

### 3.1. Impact of Laser Power

The interaction of a laser with paper changes its surface structure (Figure 1a). The paper structure becomes expanded, clearly felt under the finger, and these effects increase with increasing laser power. Simultaneously, the surface of the paper becomes slightly yellowish, and this effect also increases with an increase in the laser power applied. This effect was recorded using light reflectance measurements in the wavelength range of 400–700 nm (Figure 1b). The change in reflectance was maximum at 440 nm, and the reflectance at this wavelength was plotted as a function of the laser power used to modify the surface (inside Figure 1b). It decreased linearly with an increase in the laser power in the entire studied range.

The CIE whiteness [16] of the samples was shown to decrease exponentially with increasing laser power according to the equation y = 159.58 − 1.01 e^x/7.11^ (Figure 1c). This effect can be explained by: (i) a change in the arrangement of the fibres on the surface of the sample; or (ii) a change in the colour of the cellulose or the applied optical brighteners to yellow (although it is not visible to the naked eye) as a result of degradation under the influence of the high temperatures generated in the modification process. Additionally, the ΔE (colour difference according to the CIE Lab 1976 model; $\Delta E=\sqrt{{\left(\Delta L^{*}\right)}^{2}+{\left(\Delta a^{*}\right)}^{2}+{\left(\Delta b^{*}\right)}^{2}}$) parameter was also measured, which defines the difference between the colour of the unmodified and modified samples [14]. According to the CIE Lab, a standard observer sees the difference in colour as follows: (i) 0 < ∆*E* < 1—the observer does not notice the difference; (ii) 1 < ∆*E* < 2—only an experienced observer can notice the difference; (iii) 2 < ∆*E* < 3.5—an unexperienced observer also notices the difference, (iv) 3.5 < ∆*E* < 5—a clear difference in colour is noticed; and (v) 5 < ∆*E*—the observer notices two different colours [17]. The value of ∆*E* increases with an increase in laser power (Figure 1c), and this dependence can be described by the Boltzmann function from the family of so-called growth functions. Nevertheless, for the almost linear dependence of the ∆*E* parameter from the average laser power in the range of 12–20 W, the slope of the straight line is equal 0.362.

### 3.2. Morphology and Chemical Analysis of the Paper

The morphologies of the unmodified and laser-modified paper samples were analysed using a confocal and a scanning electron microscope. The change in the morphology of the sample surface as a result of its laser modification is also visible in the photos taken with the use of scanning electron microscopy (Figure 2). Before the laser modification, the paper was smooth, tight, and filled with some paper-enhancing substances, without the cellulose fibres protruding from the surface (Figure 2a). After the laser modification, a change in the structure of the paper is clearly visible. The photos of the modified samples show cellulose fibres coming out of the paper surface and creating roughness. In addition, the smooth layer of paper-enhancing substances is divided into smaller fragments, but no critical damages, such as cuts or burnouts, are visible in the structure. This effect (especially burnout of cellulose fibres) is stronger the higher the power of the laser used to modify the paper (Figure 2b,c). The loosening of the fibre structure is visible only on the laser-modified side. The bottom side of the paper retains its structure and does not change its filling. The action of the laser on the paper surface degrades only the paper fillers and releases the cellulose fibres from the paper structure. 

The surfaces of the unmodified and laser-modified paper samples were visualized using the confocal microscope (Figure 2). An analysis of the results showed that the surface complexity and sample height increased with the increasing average laser power used for the surface modification. The complexity of the 120 g/m^2^ grammage papers increased from 141% for the unmodified surface to 1230% after modification with maximal power (20 W).

To assess the chemical structure of the paper, the Raman spectra of the modified surfaces were collected. Figure 3a shows the spectra taken along a line perpendicular to the 20 W laser beam transition line. The intensity of the bands at the laser beam transition in the spectrum is lower than that at the unmodified site. To better visualize the differences for the unmodified surface and the surface modified with a 20 W average laser power, two representative spectra for these areas are shown in Figure 3b. The characteristic bands attributed to the cellulose are identified in the spectra at 379 cm^−1^ assigned to symmetric δ(CCC), 402 and 434 cm^−1^ assigned to ν(CCO), at 506 cm^−1^ assigned to ν(COC) of a glycosidic link, at 898 cm^−1^ assigned to symmetric ν(COC) vibration in a plane, at 969 and 997 cm^−1^ assigned to r(CH_2_) vibration, at 1091 and 1119 cm^−1^ assigned to ν(COC) vibration (symmetric and asymmetric, respectively) of a glycosidic link, at 1146 cm^−1^ attributed to asymmetric ν(CC) ring-breathing vibration, at 1292, 1338, 1379, and 1469 cm^−1^ assigned to δ(CH_2_) vibration, at 1600 cm^−1^ assigned to ν(C=C) vibration, and in region 2664–3054 cm^−1^ assigned to ν(CH_2_) [18,19,20]. The spectra shape of the unmodified and laser-modified paper is the same. The effect of a decrease in the intensity of all the bands in the spectrum for the laser-modified paper can be explained by the process of drying the cellulose fibres under the influence of the laser beam and the elevation of the cellulose fibres above the surface of the paper. In conclusion, the study did not reveal any chemical changes in the modified paper (which was only slightly affected by the laser) using the selected spectrophotometric technique. Confirmation of these findings appears to require further detailed analysis using other analytical techniques for such material in the future. 

### 3.3. Wettability of the Paper

To assess the wettability of the surface, water contact angle measurements were conducted. It can be observed that the contact angle values increased with increasing laser power and exceeded 90° for the surfaces modified with a high laser power (16 W and above) (Figure 4a). Thus, it may be concluded that a laser with a power of at least 16 W must be used for paper modification to obtain a hydrophobic surface. The hydrophobicity of the modified surfaces may have been related to the accumulation of air in the structure of protruding fibres (higher surface complexity). Simultaneously, the water drop absorption time of the paper shortened as the laser power used increased. For lower laser powers (up to 14 W), it was at least 25 min (for 10 and 12 W powers over 30 min, as indicated on the graph with an up arrow), and for laser powers from 16 W, the time was reduced to a few minutes (Figure 4b). This was probably because the spaces between the protruding fibres increased (increased surface complexity) with an increase in the laser power, and thus the water droplets passed through them easier. Additionally, the degradation of paper fillers may progress with an increase in the laser power (as described in Section 3.2), which additionally facilitates passing the water molecules through the modified surface.

### 3.4. The Unevenness Analysis of Paper Surface

The paper used in the research was designed for office use, especially for photocopying and printing text and high-quality images on inkjet and laser printers. An increased paper stiffness ensures the reliable operation of printing devices. In addition, an optimal level of whiteness ensures an eye-friendly contrast and, at the same time, guarantees a high level of contrast and colour reproduction in printing. The satin structure of the initial paper samples had a smooth surface, which guarantees the quick drying of inkjet prints while maintaining a deep shade of black and uniformly intense colours. To analyse the effect of the laser modification on their printability, the paper samples were printed with inkjet ink. The samples were then scanned and analysed for RGB to determine the non-uniformity of the prints. The obtained results were compared with the unmodified samples and are presented in Figure 5.

The RGB analysis of the samples after the printing revealed unevenness related to the printing quality. Although the printer was calibrated before work, and the print itself was made in the best quality, white stripes could be seen in the red channel, which resulted from the movement of the paper through the guide rollers. However, this effect did not further affect the analysis because the CIE Lab parameter measured with a spectrophotometer was the average value of a 2 cm^2^ area of the sample. It was shown that the distribution of the cellulose fibres released from the paper fillers was even and did not cause distortions in the colour of the samples. Additionally, a comparative analysis was performed for the RGB red channel profiles of the unmodified and laser-modified samples with 14 and 20 W (Figure 6). The RGB values for the red channel of the samples were similar to each other both for the samples before and after printing; naturally, they differed between the modified and unmodified samples. The differences between the paper samples unmodified and modified with 14 W and 20 W of laser power were approximately 31% and 30%, respectively. Changes in the RGB value referred to the change in the filling of the paper surface with cellulose fibres and did not change the intensity of the printed colour, which was also confirmed by the CIE Lab analysis (for example, the CIE Lab values for the printed sample without laser modification: *L** = 56.59; *a** = 0.78; and *b** = −41.88 and for a sample modified with a laser power of 20 W: *L** = 57.00; *a** = 1.23; and *b** = −40.09). On this basis, it was found that the change in the sample surface structure did not have a significant impact on the obtained printing effects. Moreover, the colour difference (ΔE) between the samples did not exceed 0.46, which proves that the same colour was visible on all surfaces after printing.

Light reflectance spectra were registered for the samples printed with blue inkjet ink (Figure 7a). A decrease in the reflectance value at 440 nm can be noticed for the samples modified with a higher laser power, but the dependence was not linear (inside Figure 7a).

Figure 7b shows the value of ΔE for the printed samples compared to the unprinted ones. The ΔE values for the laser-modified samples with average laser power in the range of 10–18 W were higher for the printed samples compared to the unprinted ones, and at 20 W, the colour difference between the unmodified sample and the sample modified was the same for both cases. This set of results clearly confirms that the dependence of ΔE on the average laser power should be described by the Boltzmann function. The graph (Figure 7b) shows the parameters of this function describing the experimental results, and also tangents to these curves at their inflection points are marked. These lines are not parallel to each other, but the estimated slopes of the straight lines describing the linear dependence of the ΔE parameter on the power (12–20 W for the unprinted sample and 12–18 W for the printed sample) are almost identical (0.362 and 0.365, respectively). In summary, the printing process shifted the ΔE values towards higher values until reaching the saturation value (~3.3) identified in the case under study at 20 W laser power (Figure 7b).

### 3.5. Influence of Paper Thickness on the Effects of Laser Modification

The influence of the paper thickness on the effects of the laser modification was checked. For this purpose, paper samples with grammages of 120, 190, and 250 g/m^2^ were modified with a 20 W laser in the area of 3 × 3 cm^2^. Macro photos, SEM images, and images using a confocal microscope were taken to assess the morphologies of the samples (Figure 8). The photos show the modified and unmodified parts of the samples. The results show that the surface of the 120 g/m^2^ paper was smoother and more filled than papers from the same manufacturer with grammages of 190 and 250 g/m^2^. Furthermore, after the laser modification, it could be seen that the nature of the changes was slightly different, because, in the case of thicker papers, the release of cellulose fibres from the structure of the paper was not so visible. The fibres released from the paper filler were not clearly separated from the unmodified surface. Moreover, the laser-modified surface showed undulations in its structure, in contrast to the sample with a grammage of 120 g/m^2^, where the surface had a homogeneous structure without visible concavities and convexities. These observations complement the previously obtained results concerning the analysis of the morphological structure of the paper. The complexity of the paper surface modified using the laser with the power of 20 W was the highest for 120 g/m^2^ paper and amounted to 1230%.

It is presumed that papers with grammages of 190 and 250 g/m^2^ were produced in a different way, e.g., without additional gluing and smoothing of the paper surface. Unfortunately, the exact parameters of this commercial paper making are unknown. However, the surface complexity was 708% for the laser-modified paper with a grammage of 190 g/m^2^ and 924% for the laser-modified paper with a grammage of 250 g/m^2^.

### 3.6. Influence of Paper Colouration on the Effects of Laser Processing

This work also assessed the impact of laser processing on coloured paper. For this purpose, a laser with a power of 12 W was used to treat white paper with a weight of 120 g/m^2^, printed twice on both sides with yellow, red, green, blue, and black ink. Double-printing on both sides of the paper resulted in a volume saturation of the paper with ink. For comparison, coloured cartons with a weight of 190 g/m^2^ in the same colours were laser-modified with the same laser power. The samples were processed in an area of 3 × 3 cm^2^. The laser power was reduced compared to previous studies on white paper due to the strong absorption of the laser beam by the black paper, which, at a high laser power, caused the sample to burn. The samples of coloured paper and cartons before and after the laser modification are presented in Figure 9. There was a visible reduction in the colour intensity after the laser treatment. The effect was more intense the darker the colour of the processed paper was, which is associated with an increased absorption of the laser beam. For the black carton, a slight burning of the sample was observed.

A comparison of the colour change of the printed papers and coloured cartons after modification with a laser, expressed in the CIELab system, is presented in Table 2. The values of *a** decreased after the exposition of the samples to irradiation, apart from the values for the green paper and the green and black cartoon, which increased. The values of *b** decreased, except for the values of the green and blue paper, and blue and black cartoon, which increased. A decrease in the colour intensity after the laser treatment was observed for each colour sample, expressed as an increase in the value of the L-coordinate. The decrease in the colour intensity of the samples after the laser modification was greater for darker colours, because they absorbed the laser beam more than pale colours. For the yellow and red printed papers and cartons, the colour change after the laser modification was less than 3%. As for the darker colours, the colour of the printed papers was reduced in intensity after the laser modification by 24, 28, and 17% for green, blue, and black, respectively. For coloured cartons, there was a decrease in colour intensity by 4, 11, and 7%, respectively. On the other hand, the colour intensity of the white paper decreased by 40% after laser modification with the same laser power. This effect resulted from the change in the optical properties of the paper surface after the release of cellulose fibres from the paper fillers and the high temperature of the laser beam. The change in the paper structure caused a slight change in the colour towards yellow, which resulted in a decrease in the degree of paper whiteness.

### 3.7. Example Applications

The above-presented results of the impact of laser processing on the properties of papers of different grammages and colours can find many practical applications. Figure 10 shows an example of using a pulsed laser to mark paper with microprints. The text “DosLab” of various sizes was laser engraved on the surface of the paper (Figure 10a) and visualized with an SEM microscope (Figure 10b–d). The size of the letters was measured in the VEGATG software and recalculated into the font size. The smallest letter that could be read under the microscope was 1.1 mm (Figure 10d, upper inscription), which corresponds to a font size of 3 pt. (1 pt. = 1/72 inch). These types of prints can be used as security for documents.

In the case of coloured paper characterized by a high absorbance of the wavelength of the laser used, it is also possible to create concave patterns on these papers, which can be used for marking paper products or as a decorative element on paper packaging (Figure 11).

The presented results for changing the tactile properties of the paper surface as a result of the laser modification enable the creation of Braille inscriptions on packaging or paper labels in order to facilitate product identification for people with visual impairments. Braille is one of the most common methods of social adaptation for visually impaired people. Inscriptions in Braille are embossed on information signs, elevators, ATMs, and drug packages. However, this is not a widespread practice when it comes to food packaging, cosmetics, or other everyday products. This is a major obstacle limiting the independence of visually impaired people who cannot do their own shopping in self-service stores, where there is a greater variety and the possibility of choosing goods. They have problems with navigating in the store and finding and identifying products. However, the problem does not end with leaving the store, because, when buying products of a similar shape and size, they still have a problem with distinguishing them at home [21,22]. Marking the alleys in self-service stores and packaging and product labels using the Braille alphabet would be a great help. This would significantly increase the independence and satisfaction of visually impaired people [22]. Nowadays, a great facilitation for visually impaired people and an alternative to Braille inscriptions is technology, specifically software that enables text-to-speech conversion [23], scanning devices that enable barcode scanning and product presentation from a database [24], or systems that use a camera and a database to identify text, objects, and people [22]. However, it should be noted that not every visually impaired person has the financial opportunity to purchase these types of devices. It would be much better to apply Braille inscriptions on a larger scale, which would greatly improve their quality of life without incurring additional costs.

Big problems in the dissemination of Braille inscriptions are the costs and difficulties associated with meeting the guidelines of the Braille alphabet using the most popular method of making such inscriptions, which is embossing [22]. It is considered equally problematic to use swelling printing to create Braille inscriptions due to the limited process control and fusion of dots due to the expansion characteristic of capsule paper, which may cause problems with the legibility of subtitles [25]. According to the American National Standards Institute [26], braille dot heights should be 0.019 inches (which corresponds to 0.48 mm). In Figure 2, it is shown that the treatment of 120 g/m^2^ white paper with a laser power of 20 W allowed for a maximum surface height of 0.51 mm. It is therefore believed that it is possible to select the parameters in such a way as to obtain the required height of Braille inscriptions using laser processing, and this will be the subject of further research. At this point, it should also be noted that, with the help of a laser, we can transfer the world documented in photos, thus making it easier for the visually impaired to explore the world that is not within their reach (Figure 12).

The paper laser modification process allows for the creation of decorative patterns and graphics on paper labels and can be used for food or cosmetic products to improve their design and aesthetic aspects. As another example of paper laser processing, DosLab inscriptions were made on packaging cartons: white and brown (Figure 13a,b). It is also possible to create barcodes and QR codes on paper using laser processing. For this purpose, a barcode (Figure 13c) and QR code (Figure 13d) were made on the black carton (190 g/m^2^) and white paper (120 g/m^2^), respectively. Thus, the laser technique can be used for marking paper products or marking decorative elements on paper packaging.

Laser modification can be also used to degrade optical brighteners, paints, and metallized coatings from paper in a controlled manner. Figure 14 shows photographs before and after the laser modification of paper with an optical brightener with two different laser powers (14 and 20 W). After the laser modification, no glowing of the sample under the UV lamp was observed, which proves that the brightener was partially degraded. This indicates that the optical brightener inactivation effect can be controlled by adjusting the laser power accordingly.

To sum up, laser processing, which finds numerous different applications, e.g., [27,28,29], was also shown for a modification of paper towards practical applications, such as microprinting, labelling with barcodes and QR codes, tactile detection, the controlled degradation of optical brightener, and paint and metallised coatings from paper packaging, as well as emphasizing the design and aesthetic aspects of paper packaging. 

## 4. Conclusions

The paper presented a method of paper modification with a laser. As a result of laser processing, the surface of the white paper slightly turned yellow and became rough (higher surface complexity), and this effect increased with increasing laser power. Laser modification degraded the paper fillers and released/uplifted cellulose fibres from the surface. The increase in laser power caused an increase in the value of the contact angle and, simultaneously, a reduction in the time the droplets took to be absorbed by the paper. Other effects were noted for coloured papers, which were related to the method of paper making, paper grammage, and the intensity and shade of the colour. The darker the colour of the paper, the stronger the laser beam was absorbed by the paper, and the more intense the visible change in paper colour.

The paper laser modification presented in this work can be used to mark paper with microprints, create recesses on black papers, and roughness and convexities on white papers, which can additionally be used to make Braille inscriptions on packages for visually impaired people. The use of various methods of laser modification in different combinations can also be a new form of protecting valuable products against counterfeiting at various stages of production. Such security of semi-finished products may also affect the logistics system, because labelling and marking make it possible to monitor the flow of raw materials between individual stages of product manufacturing. Additionally, laser markings on paper can be used for labelling products with barcodes and QR codes and for the controlled degradation of optical brighteners, paint, and metallised coatings from paper packaging. Moreover, they can be used as an ornament on paper packaging to increase their aesthetic value.

The developed solutions do not exhaust the subject of modifying the surface of paper with a laser and can be continued, including various paper substrates produced by various techniques.

## Figures and Tables

**Figure 1 materials-16-06691-f001:**
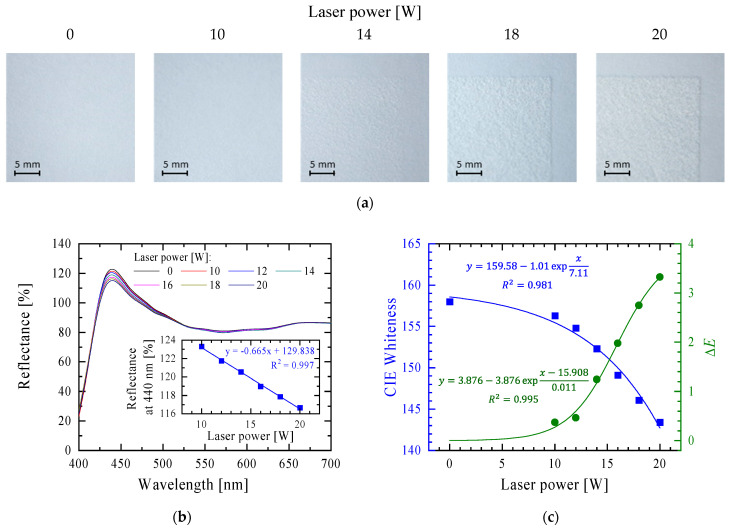
Changes in the properties of 120 g/m^2^ grammage paper after laser modification with different average laser power. (**a**) Surface structure as seen with the reflectance spectrophotometry instrument. (**b**) The reflectance spectra in the range 400–700 nm (inside, reflectance at 440 nm as a function of laser power). (**c**) The CIE whiteness and ΔE parameter of the laser-modified paper samples. The ΔE parameter is described by the Boltzmann function extrapolated to 0 W of average laser power, for which the ΔE parameter is 0.

**Figure 2 materials-16-06691-f002:**
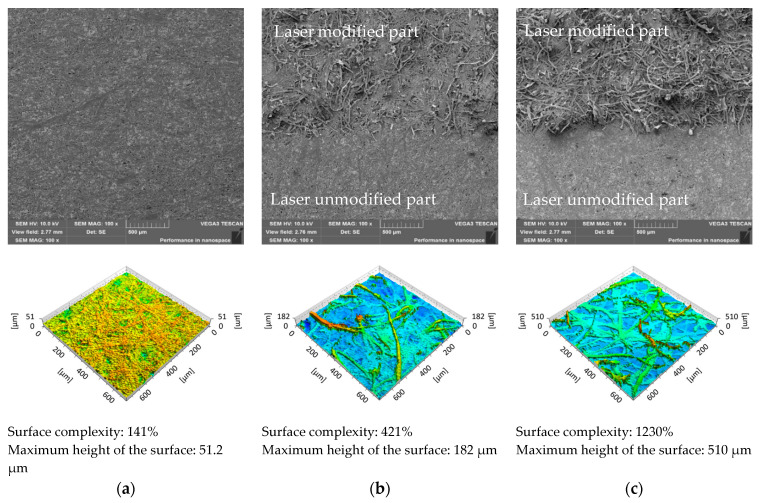
SEM images and 3D view (visualized using a confocal microscope) of 120 g/m^2^ grammage paper surface: unmodified (**a**), modified with 14 W (**b**), and 20 W (**c**) laser power. The SEM images of the modified sample contain (lower part) the unmodified part of the surface for comparison.

**Figure 3 materials-16-06691-f003:**
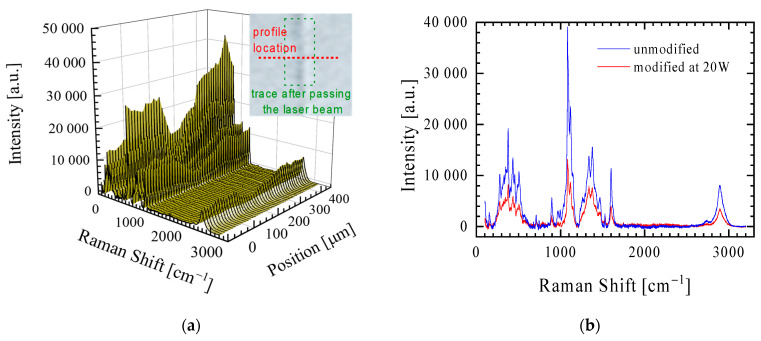
(**a**) An example of a profile of the Raman spectra of a paper surface made along a line perpendicular to the line of passage of the laser beam (see the image inside the graph with the profile marked as a red dotted line). (**b**) Raman spectra of the unmodified (profile edge) and modified with a laser with a power of 20 W (profile centre).

**Figure 4 materials-16-06691-f004:**
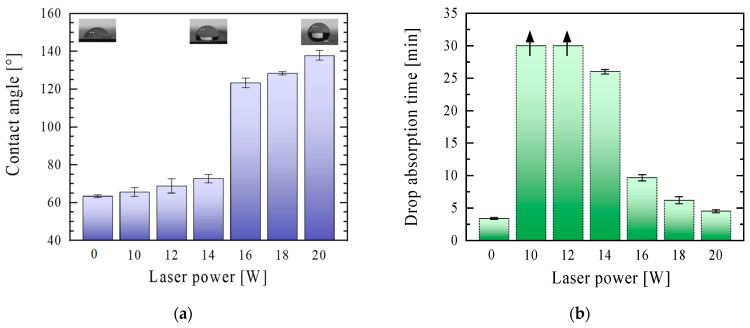
The contact angle values (**a**) and water drop absorption time (**b**) of unmodified and laser-modified paper samples with a power in the range of 10–20 W (the arrows in the graph indicate that the absorption time of a drop of water is longer than 30 min).

**Figure 5 materials-16-06691-f005:**
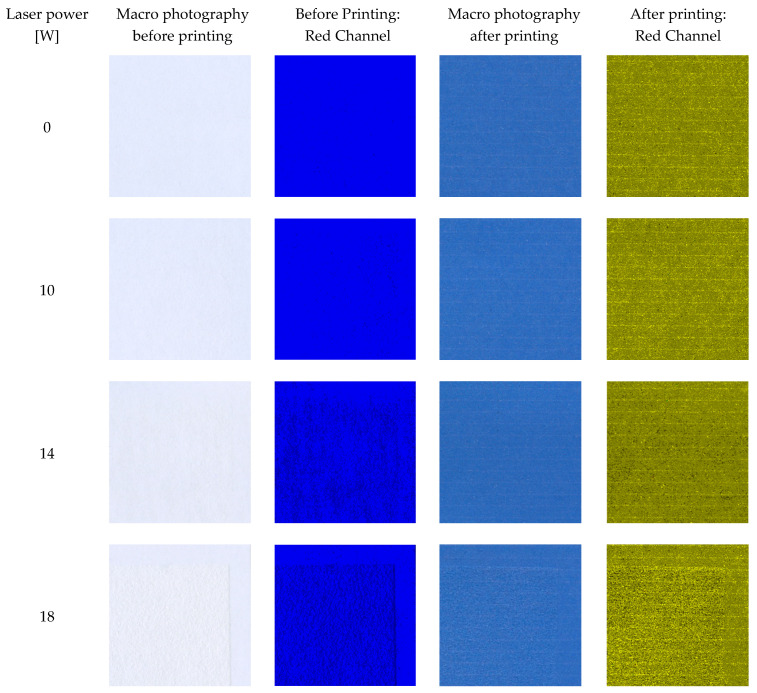
The macro photographs of paper samples before and after printing and RGBreader unevenness analysis.

**Figure 6 materials-16-06691-f006:**
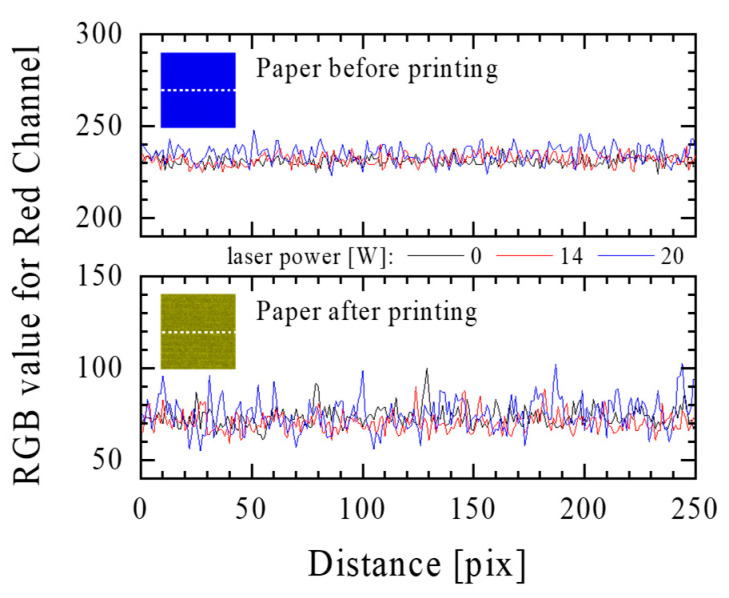
The red RGB channel profiles (1 pix = 0.1 mm) for non-printed and printed paper samples before and after laser modification with the power of 14 W and 20 W. Insets are the images of the RGBreader red channel of paper samples with white dotted lines to indicate the position of the profiles.

**Figure 7 materials-16-06691-f007:**
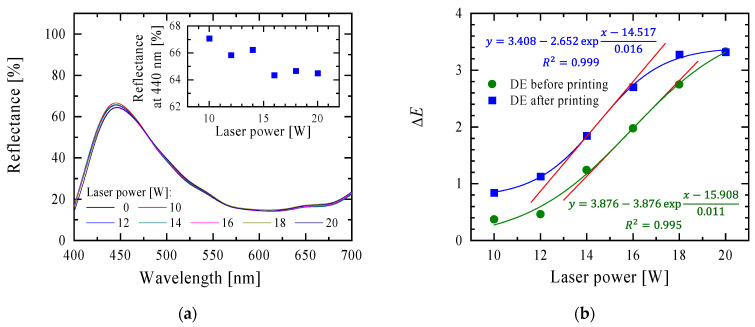
(**a**) The reflectance spectra in the range of 400–700 nm (inside, reflectance at 440 nm) of paper (120 g/m^2^ grammage) printed with blue ink using an inkjet printer, which was modified with a laser of various average powers. (**b**) The comparison of ΔE values for laser-modified samples unprinted and printed with blue inkjet ink. Tangents to these curves at their inflection points are marked with the red lines.

**Figure 8 materials-16-06691-f008:**
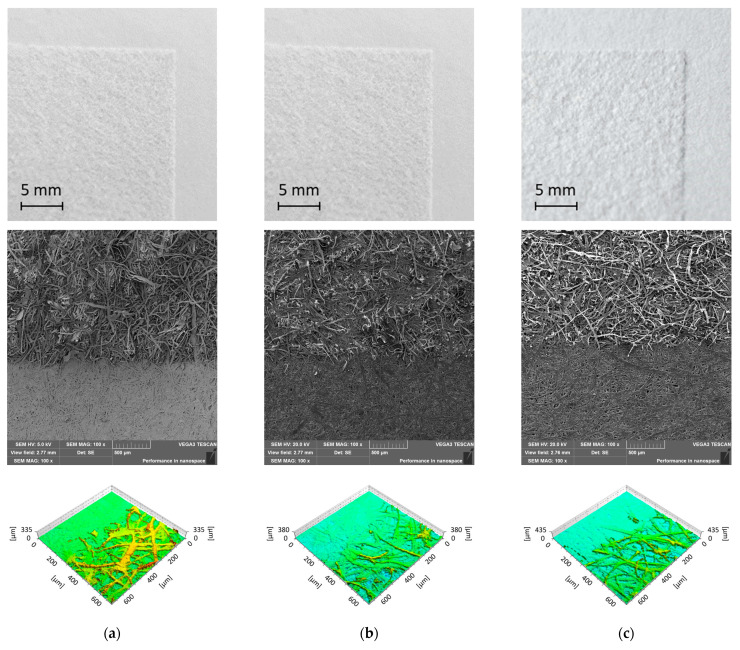
Respective from top, macro photography, SEM image, and 3D view (results from confocal microscope) of the border of the modified–unmodified part of paper samples with grammages of 120 (**a**), 190 (**b**), and 250 g/m^2^ (**c**). The modification process was carried out at 20 W average laser power.

**Figure 9 materials-16-06691-f009:**
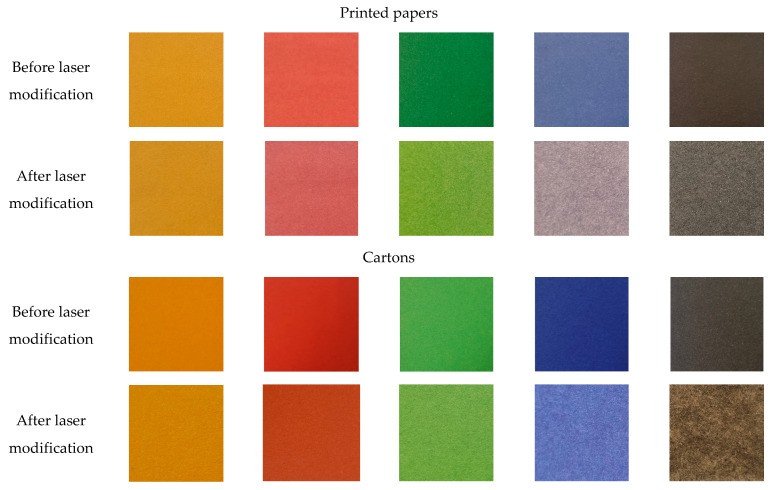
Photographs of paper and carton coloured samples before and after laser modification (12 W). All photos were taken with a Canon EOS 50D digital camera under D65 illuminance.

**Figure 10 materials-16-06691-f010:**
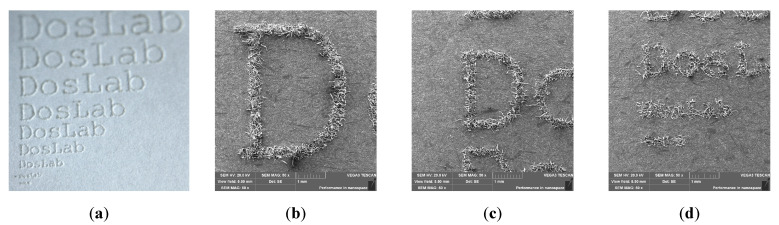
Macro photography (**a**) and SEM images (**b**–**c**) of the words Doslab made with a 20 W laser on 120 g/m^2^ paper with various font sizes: 13 pt. (**b**), 8 pt. (**c**), and 3 pt. ((**d**), upper inscription) (DosLab is a research group at the Lodz University of Technology, Poland).

**Figure 11 materials-16-06691-f011:**
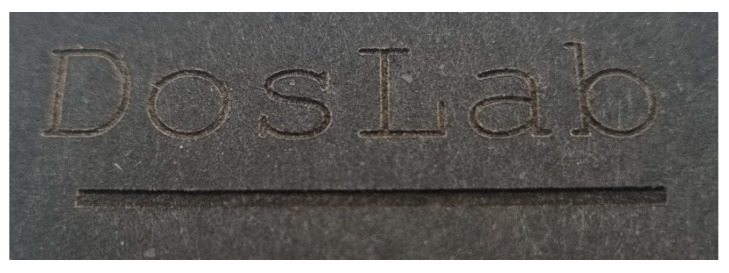
The inscription DosLab and a line creating recesses made with a laser on black paper.

**Figure 12 materials-16-06691-f012:**
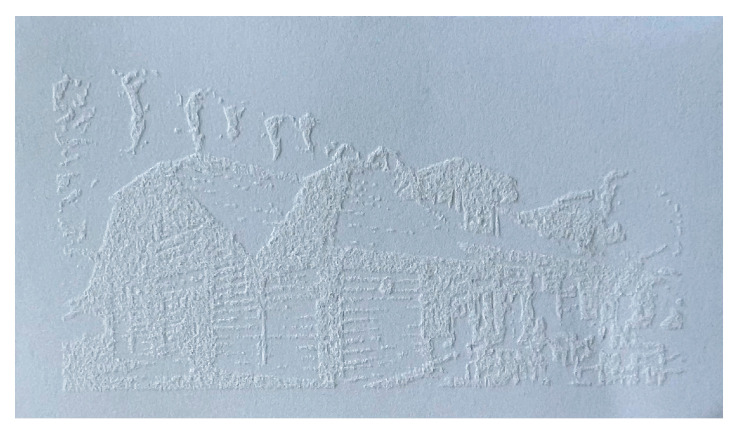
An example of graphics transferred to paper (business card size) with a laser, allowing visually impaired people to explore the world documented in photos.

**Figure 13 materials-16-06691-f013:**
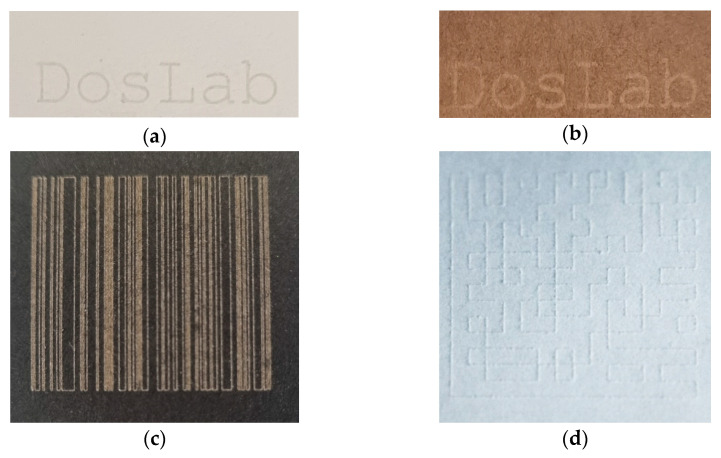
DosLab inscriptions made with a 12 W laser on white (**a**) and brown (**b**) packaging paper. Example of a macro photo of a barcode (**c**) and a QR code (**d**) made with a 20 W laser on paper.

**Figure 14 materials-16-06691-f014:**
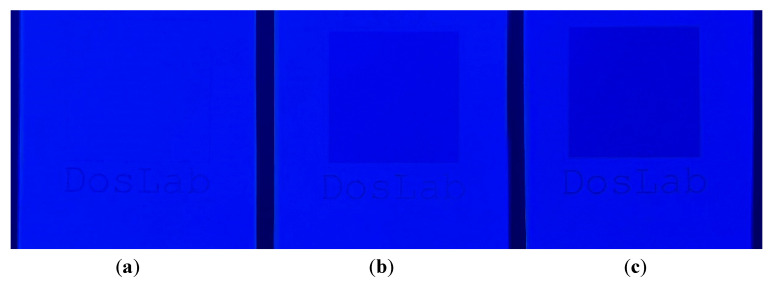
The photographs of paper under a UV lamp showing possible degradation of optical brightener from paper by laser modification: unmodified (**a**), modified with an average power 14 W (**b**), and 20 W (**c**).

**Table 1 materials-16-06691-t001:** The paper properties.

Property	Value
Grammage	120 g/m^2^
Thickness	125 µm
Whiteness in CIE system	128%
Opacity	95%
Bendtsen roughness	140 cm^3^/min
Sheet size	A3
Type of finish	Satin

**Table 2 materials-16-06691-t002:** The colour changes of coloured papers and cartons before and after laser modification with the power of 12 W. The values presented in the table are the average of 10 measurements.

Paper Colours		Before Laser Modification			After Laser Modification		
		*L**	*a**	*b**	*L**	*a**	*b**
Unprinted paper 120 g/m^2^	White	93.79 ± 0.07	3.44 ± 0.01	−15.74 ± 0.01	56.69 ± 0.06	−0.04 ± 0.00	−41.11 ± 0.04
Printed paper 120 g/m^2^	Yellow	70.08 ± 0.07	21.66 ± 0.02	54.86 ± 0.05	71.1 ± 0.07	19.96 ± 0.01	52.87 ± 0.05
	Red	54.51 ± 0.04	40.64 ± 0.04	21.78 ± 0.02	56.01 ± 0.06	38.22 ± 0.04	20.39 ± 0.01
	Green	52.66 ± 0.05	−29.52 ± 0.03	18.84 ± 0.02	65.52 ± 0.06	−23.13 ± 0.03	27.1 ± 0.03
	Blue	44.63 ± 0.04	6.97 ± 0.01	−36.75 ± 0.04	57.07 ± 0.06	5.03 ± 0.04	−16.36 ± 0.02
	Black	29.96 ± 0.02	2.22 ± 0.01	−1.81 ± 0.01	35.03 ± 0.04	2.07 ± 0.02	−3.76 ± 0.01
Coloured carton 190 g/m^2^	Yellow	67.76 ± 0.06	35.55 ± 0.04	56.70 ± 0.06	68.51 ± 0.07	35.08 ± 0.04	55.99 ± 0.06
	Red	50.98 ± 0.05	55.24 ± 0.06	31.85 ± 0.03	51.86 ± 0.05	54.86 ± 0.05	31.66 ± 0.03
	Green	69.83 ± 0.06	−34.49 ± 0.04	41.9 ± 0.04	72.52 ± 0.07	−31.43 ± 0.03	40.95 ± 0.04
	Blue	40.89 ± 0.04	9.38 ± 0.01	−42.04 ± 0.05	45.57 ± 0.05	5.35 ± 0.05	−38.07 ± 0.04
	Black	28.41 ± 0.03	1.17 ± 0.01	−0.41 ± 0.00	30.37 ± 0.02	2.42 ± 0.02	9.59 ± 0.02

## Data Availability

Data available on reasonable request by contacting the corresponding authors.

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
