# Peer review of "The Surface Modification of Papers Using Laser Processing towards Applications"

_materials, 2023, doi:10.3390/ma16206691_

Round 1

Reviewer 1 Report

1.     Add scale bar in Figure 8.

2.     In Figure 9, are they real color pictures?

3.     Can authors add more discussions on the potential of this work?

4.     It is better adding more descriptions on the advantages of laser fabrication. The following references maybe helpful, such as FRONTIERS IN CHEMISTRY 2019, 7, 461; ADVANCED MATERIALS 2020, 32, 1901981; PHOTONICS RESEARCH 2020, 8, 577; FRONTIERS IN CHEMISTRY 2022, 9, 823715; FRONTIERS IN CHEMISTRY 2022, 10, 1073473.

Reviewer 2 Report

The authors investigated the surface modification of papers through laser processing. This is an interesting topic. I would like to see the publication of this paper after modification. The possible application of this research is recommanded to be clarified in the Introduction section, instead of the Conclusion section. The English of the paper is also recommanded to be polished to make it better.

Reviewer 3 Report

The paper entitled “Surface modification of papers using Laser Processing towards Applications”, aims to show the influence of laser processing on paper properties. But based on the information written in this research, it can be said that the authors know or have presented a  lack of knowledge about the paper properties and composition. In addition, there are few information in the paper that are completely wrong. The paper lacks in scientific contribution. This means that there are no scientific explanations of the results. The authors only have compared the results.

The Englis in paper should be enhanced- for example (line 78) “the printing of commercial paper”

Lines 43-45: I sprinting only used for the production of banknotes? Do phosphorescent inks have any similarities to laser printing? For which printing application are leser used the most, and for which materials?

Lines 53-54: is only a reduction of cellulose molecular weight a result of laser printing? I don’t think so.

What about the production of CO2  during laser modification of paper and paperboard?

lines 77-80: For better clarity, the paper properties should be summarized in one Table

lines 94-95:  since the chemical composition of inks can be a crucial in this, a strictly defined method of coloration and properties of inks used for the coloration of cartons needs to be specified.

Line 98: what mean 10 in D65/10. Missing something here, please add

Line 100: missing measuring step in nm

Line 135: isn’t the 300  dpi a too low resolution for the evaluation of unevenness?

Line 141: hod did you determine the threshold?

Line 155: why does the CIE whiteness decreases with the increase of laser power, for what reason? What is destroyed in paper?

Line 157: what is DE parameter, how it was determined? What does the DE values say about the sample?  In addition, the abbreviation DE is written incorrectly

Lines 158 -159: “A clear difference in colour to the naked eye is noticed for DE ≥ 0.46.” – this statement is not correct and completely wrong

Lines 175-176: “Before laser modification, the paper is smooth, tight and filled with some paper enhancing substances without the cellulose fibres protruding from the surface” -  you haven't measured surface smoothness and therefore you cannot talk about the smoothness of the paper and you cannot claim that before the laser modification the paper was smooth. Why haven't you measured the smoothness or the roughness of the paper? (lines 304-305)

Lines 180-182: can you claim that there was no burnouts or mechanical damages of the fibres? To which results you can compare yours and claim this?

Lines 91-192: Does it mean from 141% for unmodified surface to 1230% after modification? How was this determined?

Lines 197-214: The Raman spectroscopy is commonly used for the evaluation of. Inorganic particles in paper. The FTIR spectroscopy is commonly used for the evaluation of chemical composition of paper in the terms of organic substances. Therefore, this Raman spectroscopy is not sufficient method for the evaluation of chemical composition of paper. In addition, the paragraph about the Roman spectroscopy is written minimalistic and needs to be expanded

lines 214 – 215: Figure 3a is unclear and overlapping of measurements does not provide a clear image. Some improvements needs to be made in order to have a clear picture and to support described results. In this form is not good.

Line 220 : method was used for the evaluation of the water contact angle? LaPlace / Young?

In Figure4b what do arrows means?

Line 245: missing the printing process data (type of printer and printing ink) in experimental part

Figure 9: Are the obtained photographs results of  samples scanning as described in section 2.7? if not, please describe the method. From the Figure caption I can not clearly understand what method for the evaluation was used.

Lines 285-294 : about what DE are you talking about?

Line 358: Colorimetric parameters should be written according to CIE LAB system in italic with a * (L* a* and b*). In addition, please add in the experimental part how colour was measured. (check the whole manuscript before and after the mentioned lines)

Line 357: missing the standard deviation of colorimetric parameters in Table 1. In addition, how do you comment on the results in table 1. What do number say?

Figure 14 does not show the removal of removal of the optical brighteners, but only its degradation. The paper is still fluorescent under the UV lamp.

Line 447: what does it mean “releases cellulose fibres from the surface”?

The Englis in paper should be enhanced- for example (line 78) “the printing of commercial paper” 

the style and grammar errors

Round 2

Reviewer 3 Report

Authors are encouraged to make changes as required in the first revision. Most of the comments were ignored and rejected by the authors. Some of the authors comments show the absence of basic knowledge in the field of printing, color management and materials properties.

The additional table or figure in the text won't make a problem in understanding the text oor meaning, but opposite. The summarized table would give a better presentation of the results. Thus, please add the summarized table regarding the paper properties. ( lines 79-82)

The question about the differences of phosphorescent inks and laser printing was just to make a clear that they are different, so I don't see the point that you refer to different technologies and inks. It would be better to justify the printing processes with the theory and refrences similar to your research.

line 104: there is no need for additional explanation of D65/10 but just to add a degree sign in order to make a correct form, thus please correct it.

line 95: what do you mean by "Various colours" - which one? magenta, cyan, red, or what?

line 105: it is not 10 nm resolution, but 10 nm step.

please write down the used CIEDE formula used in this study.

line: 175-175: ∆??? ∗ (∆??? ∗ = 1 is at the threshold of human perception) - this statmenet is incorrect. Please add the refrence you used for this statement, and explore the correct meaning. The DE=1 cannot be percieved by the human eye

The Figure 3a is still unclear. The comments on Raman vs. FTIR is not that you have found the refrence that justfy your results. The refrence you have listed are from the 1987, 1994 and 1997 year? Isn't that clear enough? It is simply not possible that raman spectra remain the same after laser modification. Raman and FTIR spectra show even the smallest changes ocuuring in the paper properties due to degradation, or even due to printing. Thus the whole setion need an upgrade.

During the paper production, the paper several different processes. During my stdy of paper, I have never find the literature that claims that cellulose fibers are above the surface of paper. That is simply not true. Have you ever made a cross section of paper and have you ever seen a cellulose fiber outside the paper surface?

Standard deviation is rquired when presenting CIELAB parameters. the 4% in colorimetric measurements id quite big.

Gramatical and style errors
